# Social networks influence farming practices and agrarian sustainability

**Amaia Albizua**[1,2]*, **Elena M. Bennett**[2], **Guillaume Larocque**[3], **Robert W. Krause**[4¤], **Unai Pascual**[1,5,6]

**1** Basque Centre for Climate Change, Leioa, Spain, **2** Department of Natural Resource Sciences, McGill University, Ste. Anne-de-Bellevue, Québec, Canada, **3** Quebec Centre for Biodiversity Science, McGill University, Montréal, Québec, Canada, **4** Institute for Analytical Sociology, The Institute for Analytical Sociology, Linköping University, Norrköping, Sweden, **5** Ikerbasque Foundation for Science, Bilbao, Spain, **6** Centre for Environment and Development, University of Bern, Bern, Switzerland

¤ Current address: Methods and Evaluation/Quality Assurance, Department of Education and Psychology, Freie Universität, Berlin, Germany

* amaiabc3@gmail.com

## Abstract

The social-ecological effects of agricultural intensification are complex. We explore farmers' perceptions about the impacts of their land management and the impact of social information flows on their management through a case study in a farming community in Navarra, Spain, that is undergoing agricultural intensification due to adoption of large scale irrigation. We found that modern technology adopters are aware that their management practices often have negative social-ecological implications; by contrast, more traditional farmers tend to recognize their positive impacts on non-material benefits such as those linked with traditions and traditional knowledge, and climate regulation. We found that farmers' awareness about nature contributions to people co-production and their land management decisions determine, in part, the structure of the social networks among the farming community. Since modern farmers are at the core of the social network, they are better able to control the information flow within the community. This has important implications, such as the fact that the traditional farmers, who are more aware of their impacts on the environment, rely on information controlled by more intensive modern farmers, potentially jeopardizing sustainable practices in this region. We suggest that this might be counteracted by helping traditional farmers obtain information tailored to their practices from outside the social network.

## Introduction

Agriculture is undergoing a transformation, mainly through intensification, worldwide [1]. This is reflected in the increased numbers of industry-oriented farms, characterized by large-scale monocultures with high use of pesticides and fertilizers and, which often go hand in hand with investments into irrigation technology. Globally, this intensification process is driven by a push for higher yields [2] and the allocation of increased amounts of food crop production toward biofuel production to meet energy demand [3]. This agricultural

**Data Availability Statement:** All relevant data are within the manuscript and its Supporting information files.

**Funding:** AA, Eusko Jaurlaritza(Postdoctoral Scholarship); https://www.euskadi.eus/

informacion/ayudas-al-personal-investigador-programa-posdoctoral/web01-a2hunib/es/; Ministerio de Economía y Competitividad(ES) (MDM-2017-0714). EB, Canadian Network for Research and Innovation in Machining Technology, Natural Sciences and EngineeringResearch Council of Canada RK, European Research Council (ERC) under the European Union's Horizon 2020 research and innovation program (grant agreement No. 648693).

**Competing interests:** The authors have declared that no competing interests exist.

intensification is a major driver of biodiversity loss [4], which ultimately reduces non-food ecological benefits at the landscape scale, including pollination services [5], regulation of pests [6], soil quality [7], water quality [8] and cultural benefits, such as sense of place [9, 10]. Moreover, intensification of agrarian systems also alters institutions (norms and rules) and social relations in farming communities [11].

Here we adopt the nature's contributions to people (NCP) framework of IPBES [12, 13] in the context of agriculture [14]. There are several studies about farmers' perceptions and values regarding NCP [15–17] or about how farmers influence one another in their land management activities and perceptions [12, 18, 19]. However, farmers' awareness about their co-production of NCP and what affects farmers' interactions when they make land management decisions (i.e. the way such a rural network structure emerges) is not well known [20, 21]. Farmers' awareness of their potential to influence multiple NCP is crucial since it represents the first step for transformative change [4]. Moreover, most of the negative NCP due to agricultural intensification are not directly addressable by either central governments or by individuals because they emerge from the interdependent actions of many decision-makers, at different levels [22]. For example, if a given farmer aims to improve the status of underground water, she can opt to use fewer or no fertilizers or reduce pesticide use. However, if neighboring farmers do not follow suit, the impacts of one individual's changes would likely be minimal. It is also difficult to be addressable by a central government because top-down technical approaches frequently fail to build on the local knowledge, innovative capacity, and expertise of farmers of rural communities [23].

Farmers' social networks can address the environmental problems of agriculture by expanding the number of farmers using beneficial land management practices through practices such as information sharing [24]. Farmers are embedded in social networks through which information and resources are exchanged with other farmers and organizations [25]. These multi-scale interactions can allow co-operative actions and dissemination of the "know-how" between rural community members, ultimately contributing to spread of the dominant management practices in the network [24]. The influence of farmers' networks on management has been studied previously [26, 27]. Some studies, e.g. [28] have found that converting land to crop production was correlated to the addition of ties in a local producer's network, while the diversity of land use types was correlated with the number of institutional ties. Through the comparison of knowledge and resource exchange multi-level network structures [25], discovered that access to knowledge and resources put intensive farmers in a privileged position of power allowing to influence other farmers' land use decisions and pushing out small farmers who practice more sustainable forms of agriculture.

Yet, understanding how patterns of ties, and thus social networks, emerge is missing and crucial. To deal with social network creation understanding, we need to understand that there are three comprehensive drivers influencing the formation of social ties [29]: (1) network structure; (2) nodal attributes, and (3) external contextual factors. Network structure refers to endogenous network processes, like reciprocity (you sharing knowledge with me, makes me more likely to also share knowledge with you), transitivity (I ask those for advice that my advisors also ask for advice), or centralization (most people ask you for advice). Nodal attributes refer to the farmers' characteristics, such as their perceptions, age, gender, etc. When social tie formation depends on attributes of the nodes and how these attributes interact with the network structure (e.g., differences in network activity) and with the attributes of other nodes (e.g., homophily) we talk about nodal attribute drivers [29, 30]. Finally, external contextual factors are factors that do not depend on farmers, such as the lithology of the land they cultivate, or the distance between plots.

With the goal of linking whether farmers' awareness regarding their NCP co-production is part of the main factors for social networks creation, we develop two main objectives. Our first objective is to explore farmers' views on their role in increasing or decreasing NCP through their farming practices. We hypothesize that farmers' perceptions about NCP are aligned with their land management practices and, thus, with the outcomes of their management in terms of co-production of certain NCP over others. An example could be that farmers practicing intensive agriculture are aware of obtaining higher yields (material NCP) but they may also be aware of deteriorating traditional knowledge and landscape aesthetics (non-material NCP) so that this awareness and acceptance of such trade-offs between NCP lead to agriculture intensification spread. Our second objective is to examine to what extent structurally driven relationship formation influence the connections among farmers ultimately affecting their land management decisions. Hence, our second hypothesis is that farmers sharing similar land management practices and a similar awareness about NCP co-production are significant factors for creating the ties within farmers' social network.

We use a case study and data from a farming community in a village in the region of Navarre, Spain. The agricultural practices in this village are undergoing a major intensification process, which affects farmers' informal networks as well as their awareness of the impact of their management on NCP [11, 17, 31]. We interpret variations in farmers' engagement with the agrarian ecosystem [32] to ultimately aim at understanding the ecological consequences of their land management decisions [33]. This is especially relevant in a context where farmers' engagement with sustainability is still very low [34, 35] even if there are increasing resources devoted to rural sustainability (e.g., via the European Common Agricultural Policies).

## Materials and methods

### Study area

We studied a farming community from a village of the Navarre region in Spain (Fig 1). The name of the village will remain anonymous to protect the identity of the participants in the study. This village is located in the *Zona Media* and *Ribera Alta* zones of the Ebro River watershed, which hold 22.5% of Navarre's population [36], and which has a Mediterranean climate and an arid and semi-arid climates, respectively (as per the Papadakis classification). This village underwent agricultural intensification through the adoption of large-scale irrigation—i.e. the Itoiz-Canal de Navarra project—that began in 2006 and it is still undergoing, having converted 22,445 ha to modern irrigation across twenty-two villages in its first Phase. The Navarre government, coordinated with other Spanish administrations and European strategies, has provided farmers with infrastructure and public subsidies to favour the adoption of modern irrigation as a means to deal with current rural development challenges (e.g., productivity losses and climate variability). This has affected farmers' land management and perceptions about their practices' impacts.

The adoption of large-scale irrigation has led to several changes in land management by farmers [25]. There has been an increase in the size of the cropped land per farmer through a land re-parcelling process, since the minimal arable land in this modern irrigated system is now 5 Ha [17, 11, 31]. Before the irrigation development project there used to be many small-scale farmers laboring small plots (<1 ha) of vegetables and woody crops such as olive and almond trees, often under traditional irrigation systems [11]. Farmers who owned land in the areas affected by the modern irrigation project had to choose among three options: they could adopt modern irrigation, partnering with other farmers if they own less than five hectares; switch to lands in other areas with rainfed systems; or sell or rent out their land [11, 37]. Farmers adopting modern irrigation have introduced new crops such as corn (Zea mays), forage, or

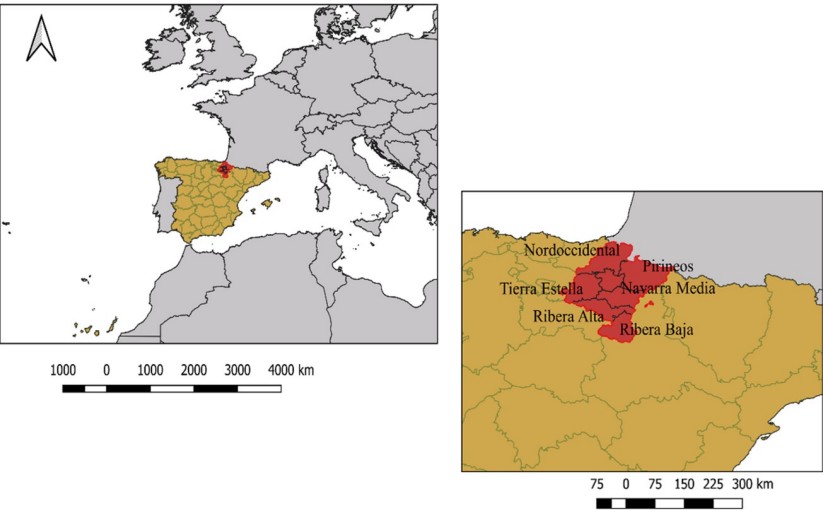

**Fig 1. Location of Navarre province, Spain.**

biofuels plantations [11]. Now many farmers employ pressure sprinklers and use higher doses of mineral fertilizers and pesticides. As a result of agricultural intensification in this region higher nitrate pollution of land and water, both in rivers and underground water has occurred [38], among other negative environmental impacts [39]. Likewise, alternative-farming practices such as small-scale and organic farming are being rapidly displaced [11].

Land and labor allocation dynamics have also changed in the region. In this regard, local cooperatives, after the large-scale irrigation project installation, and because of land concentration, coordinate farmers and landholders' land and labor exchanges [25]. Many older farmers not willing to invest in the new technology have abandoned farming and they often rent their lands to other farmers [11]. Normally, land management is allocated to younger farmers who are more prone to adapt farming to new technologies, including irrigation. Many landholders have also left their land to the agricultural cooperatives that distribute the land among active farmers for cultivation. Moreover, new kinds of cooperatives, such as those for ownership of machinery, have been created to share new machinery by farmers in the region.

## Data and methods

**Sampling for the social network analysis.** We used a list of all landowners and farmers affected by the irrigation project, provided to us by the irrigation community, and from key local informants (e.g. agrarian cooperative workers) to identify members of the farming community. We then used snowball sampling until no additional farmers or landholders' names were generated.

We got a response rate of 77% through the combination of the list and the snowball sampling (81 farmers and landowners out of the 106 people approached). The diversity of farmers was purposely searched (see [17]). The surveyed farmers, when asked about their network contacts, nominated 80 additional people (i.e. farmers and other rural community members such as rural advisors, cooperative workers, and sellers) leading to a total network of 161 people. However, only the originally approached farmers that responded to our survey (N = 81) are included in the Exponential Random Graph Model (ERGM) (explained below). Only these respondents provided information about their NCP perceptions and management behavior, which are crucial factors of our analysis.

The McGill University Faculty of Agriculture and Environmental Sciences Research Ethics Board reviewed and approved this project by delegated review in accordance with the requirements of the McGill University Policy on the Ethical Conduct of Research Involving Human Participants and the Tri-Council Policy Statement: Ethical Conduct For Research Involving Humans. Consent was informed and obtained verbally. The study did not include minors.

**Semi-structured interviews.** We collected data via in-person semi-structured interviews with farmers and landholders during June-August 2017. Through the interviews, we compiled information regarding the type of management they performed, as well as their perception regarding which NCP they co-produced through their land management and farming techniques. Also, a name generator with a free recall was used [see e.g., 22]. We asked farmers to mention up to five people [as in 43] who they considered to influence their land management decision-making. We later asked about their perceptions regarding which NCP they perceived they were impacting upon through their land management and farming choices. We asked respondents both with whom they shared knowledge and from whom they received knowledge useful for their farming practices. We asked openly about the knowledge exchange, which helped to make a list of possible topics they talked about within the farming community. We performed 32 additional interviews with organizations related to the local agrarian sector (the surveyed farmers had mentioned 27 of the organizations interviewed). To understand how other organizations can influence farmers' land management decision-making, five other organizations were selected based on a literature review and the experience of the lead author who has been researching in the area since 2013. This additional information is considered relevant to better understand the context and to triangulate the information so that we can make a better interpretation of the results.

**Semi-structured interviews qualitative analysis.** All interviews were recorded in audio-only format and transcribed only the statements associated with farmers' views about their awareness about how they co-produced NCP through their land management. These statements were additionally organized according to whether they were positive or negative, the scale at which NCP were delivered, and whether the contributions were self or other-oriented. Key parts from farmers' and organizations' interviews that were relevant to better understand results were identified (and showed in quotes below). 'Few', 'some', 'many' and 'most' are used consistently to mean less than 25 per cent, up to 50 per cent, up to 74 per cent and 75 per cent or more of the corresponding sample, respectively. This was useful to conduct a narrative analysis—i.e. making sense of our interview respondents' individual stories to highlight important aspects of their stories that will best resonate with the results.

Each question was split in categories being able to convert some of the qualitative information into measurable data. Then, we mapped connections in the data to those specific categories through statistical analysis such as hierarchical cluster analysis.

**Hierarchical cluster analysis to understand rural community composition.** We performed two hierarchical cluster analysis (HCA) to classify farmers and landholders into three groups representing different ways of land management and, different NCP co-production awareness. We included the data from the 81 farmers and the groups were created according to 1) their land management decision-making (Management Cluster- MC) and, 2) their perceptions regarding which NCP they perceived they impacted on with their land management (NCP Cluster—NCPC). For the Management Cluster (MC) the variables included were: the type of fertilizers used (organic, mineral, mixed), the type of irrigation performed (sprinkler or drip), the surface of the cultivated lands (Ha) and the crops grown (cereals, maize, grass, vineyard, energetic crops or others (including vegetables and fruit trees). Grouping farmers concerning their farming practices allowed identifying, to some limited extent, their engagement with the landscape. For the NCP Cluster (NCPC), 14 variables were included representing

different contributions they could enhance through their land management. Such NCP had previously been identified by farmers in the region [17] where socio-cultural values about NCP were identified (the NCP values include as the provision of food, biodiversity, land fertility, habitat, water regulation, climate regulation, pests regulation, land pollution absorption and soil erosion; as well as relational values such as education, traditions, landscape aesthetics, traditional knowledge and personal recreation). See S1 Table to understand the interpretation of the performed HCAs.

**Analysis of differences among farmers' groups in terms of awareness of NCP co-production.** First, we explored if there were significant differences in terms of land management among the three groups of farmers given their differentiated NCP co-production awareness. Second, we examined the NCP co-production awareness among the three groups of farmers with different land management strategies using a Kruskal-Wallis test [40–42]. A False Discovery Rate (FDR) control approach was used to counteract the problem of multiple comparisons [43].

**Selection of drivers to understand social tie formation.** To understand the role of farmers' network structures on land management decision-making, we used a mixed exploratory and hypothesis-driven approach for testing the following structurally driven tie formation factors among farmers in the network: reciprocity, transitivity, network activity, and network popularity [30, 44]. Transitivity, in this context, means that advisors of advisors are also sought for advice. Network activity and popularity refer to the number of outgoing ties (i.e, the number of farmers asked for knowledge on farming practices from a given farmer), and incoming ties (the number of farmers seeking knowledge on farming practices from that given farmer). These indicators show the presence of leaders in the farming community, who are consulted farming advice more frequently due to their social position as well as farming experience and knowledge. Leaders can bring people together [45] and may influence others' values [18, 19, 46].

We also included attribute driven factors influenced by farmers' characteristics. Homophily is the tendency for actors to seek information from those who share the same characteristics. We tested for homophily in social variables such as age, NCP awareness, and land management to see if those factors influenced the network configuration. Sharing information with others that practiced similar farming strategies is recognized as a probable practice [47].

Regarding biophysical contextual factors, we included geographic proximity in the model, which has been shown to be a major contributor to network formation (see e.g., [44]). The presence of an agricultural plot within 1 km was selected as the distance metric, with the expectation that neighbouring farmers, i.e., located nearby, are more likely to be asked for knowledge on farming practices than distant farmers. See S2 Table for a more detailed justification of why other geographic and ecological environment-related variables could not be included in the model.

**Exponential Random Graph Model (ERGM).** We used Exponential Random Graph Model (ERGM) for the analysis of the advice network. ERGMs model the formation of network ties by comparing the observed network with randomly generated networks on specific counts of network substructures (e.g., the number of ties in the network, the number of reciprocated ties, the number of triangles, the number of ties within and between certain groups etc.). The parameters of the ERGM each correspond to one of these substructure counts with a positive parameter meaning that the respective structure (e.g., number of reciprocated ties) is occurring more often than expected by chance (given the rest of the model), while a negative parameter means that the respective structure is occurring less often than expected by chance (given the rest of the model). In our model, we included parameters for the substructures of interest, for instance, the number of ties within the different land management types. A

positive parameter here would indicate that farmers are more likely to ask those farmers for advice that share similar land management styles, while also controlling for, among others, the tendencies that farmers ask those for advice that they ask themselves (reciprocity), that they ask those for advice that their advisors also ask for advice (transitivity), and that they might ask those farmers that manage land close to them or those of similar age. Network structures are highly dependent on each other and thus changing one structure (e.g., increasing the number of reciprocated ties) also changes the frequency of other network structures (increasing reciprocated ties also increases transitivity), hence complex network models like ERGMs are required to properly model network processes and take their interdependencies into account. For a detailed introduction into ERGMs see [50].

The ERGM was used to test whether land management related knowledge relationships are associated with a similarity of farming types, social characteristics, and geographical proximity. ERGMs permit testing all of these variables simultaneously and predicting edge formation [48], therefore informing about how land management related rural networks are created. We fit the ERGMs using the ergm (v2018.10, Handcock and Gile, 2010; Handcock et al., 2014) package for R statistical software (R version 3.6.3) programming language (R Team et al., 2020). We constrained the simulations to a maximum of 5 nominees. Sometimes, especially when the number of configurations included is high or the models are not well specified, ERGMs do not converge [29]. To deal with this we started with a simple model and added more configurations incrementally if there were reasons to believe they were important (see S2 Table regarding our reasoning to include environmental variables). After obtaining a structural baseline model, we first tested each new set of parameters, added to the simple, smaller model, and then later tested a final model with all the previously significant parameters. This way we can explore the data in several directions and preselect relevant effects while reducing the number of false positives by testing all relevant effects jointly in a final model.

**Missing data.** Generally, farmers answered about awareness of their co-production of NCP. However, some of the landholders who did not manage land anymore did not answer this question, and, for this reason, we took a subsample of 55 farmers regarding the analysis about their perceptions of NCP co-production. As for the ERGMs analysis, this can be reliably estimated under missing data (assuming the data is missing at random), as long as the missing data is only on the network ties [49]. Missing covariates, however, cannot easily be handled within the ERGM framework because the model is only generative regarding network links and treats all covariates as exogenous (this means that missing covariate values cannot be internally imputed during the estimation). We thus base our findings for the ERGM only on a subsample of 80 respondents on which we have nearly complete information. One farmer was the only one who used purely organic farming techniques and asked no-one else for knowledge on farming practices, nor did anyone ask this farmer for such a knowledge. This farmer thus represented a very special case and was excluded. It has been shown that ERGMs still provide reliable results with 20–30% missing data [50, 51].

## Results

### Farmers' awareness about their role in NCP co-production

We found different levels of awareness among farmers regarding how they can affect NCP within the agricultural landscape given their land management decisions. Not surprisingly, the provision of food was the most highly cited material contribution of farmers to NCP. Around 96% of the farmers answering this question thought they contributed to the provision of food through their crop production. This was followed by some cultural contributions, such as impact on their own recreation (87%) and farmers' influence on landscape aesthetics (80%).

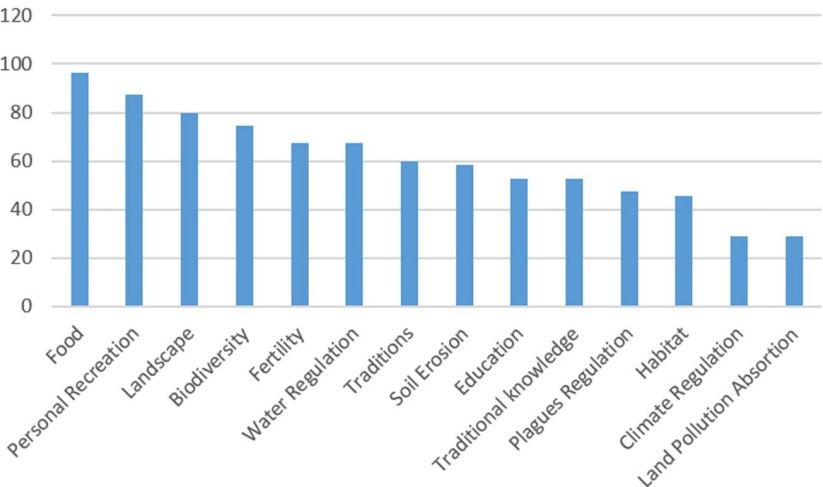

**Fig 2. Percentage of farmers aware of their NCP perceived as co-produced through their farming.**

Farmers we interviewed also mentioned perceptions about their impacts in creating an adequate habitat to enhance biodiversity (75%) and to support water regulation (67%). In contrast, their impact on land pollution and climate regulation was the least cited, being mentioned by only 29% of farmers (See Fig 2).

## Differentiated groups of farmers and their differences regarding management and NCP co-production awareness

The result of the first HCA grouped farmers according to their land management (MC) into three groups: modern technology adopters (N = 45), traditional farmers (N = 21), and landowners (N = 15). The second HCA was based on their awareness of their role to co-produce NCP (NCPC) and it also revealed three different groups: farmers conscious of their negative impacts on non-material NCP (N = 19); farmers aware of their contribution to climate regulation and landscape aesthetics (N = 42); and farmers more aware of their role enhancing the habitat conditions, maintaining cultural traditions and traditional knowledge, and regulation of pests (N = 20). See Table 1 for more detail on the characteristic of each group.

We did not find any significant differences (after adjusting for multiple comparisons) among the farmers performing different kinds of land management in their awareness of NCP co-production. However, we did find some indications that some land management decisions, such as the type of irrigation and fertilizer use, crop selection, and, to a lesser extent, the area of land farmed, are correlated with the NCP co-production awareness (see Fig 3). Those using drip irrigation and who normally cultivated "other" crops encompassing wooden fruit trees and vegetables were normally aware of their contribution to climate regulation, landscape aesthetics, traditions, and habitat. Interviews revealed that some traditional farmers considered planting fruit trees a way to regulate the climate and to regulate land pollution. They explained that trees had higher capacity to absorb carbon and retain pollutants, in comparison to other crops. Likewise, they attributed the improvement of the landscape aesthetics to the trees they grew. Traditional farmers also normally linked the way they pruned the vineyards or the olive trees to keeping traditional knowledge and traditions.

In contrast, many modern irrigation adopters emphasized that such implementation had largely changed their management and they believed they were not contributing to traditions

**Table 1. Network nodes characteristics in the two hierarchical cluster analysis (Based on 81 community participants).**

| HCA1 based on management | HCA2 based on NCP co-production awareness |
|---|---|
| *Modern farmers* (N = 45) hold large areas with agro-industry oriented crops (maize, grass, biofuels, cereals), sprinkling irrigation, mixed fertilization They make all decisions about farming and investments. They normally contract labor. | *Conscious of their negative impacts* (N = 19) These farmers realize their practices enhance soil erosion, land pollution, and habitat degradation. |
| *Traditional farmers* (N = 21) hold plots between 0–5 hectares of "other" crops (vegetables, fruit trees such as olive or almond trees). Hardly any sprinkling irrigation, not main commercial crops (maize, vineyards, and biofuels). Mixed variety of fertilizers. Normally small-scale farmers do not rely only on agriculture, but have other sources of income or are retired farmers. | *Climate regulation and landscape advocates* (N = 42). These farmers consider they helped on climate change mitigation through the store of carbon underground and above ground in more perennial crops such as wooden crops. Likewise, they are aware that their farming practices, types of crops, and rotations contribute to improving landscape aesthetics. |
| *Landholders* (N = 15) are not directly associated with farming and they do not make decisions on it either on technology investments. They are normally retired farmers who own many small surfaces of arable land in the village and rent those lands to other farmers to labor them. | *Traditionalists and habitat supporters* These farmers enhance regulating and cultural contributions (N = 20) Aware of their role to enhance the habitat conditions, traditional knowledge, traditions maintenance, and regulation of pests. |

and traditional knowledge anymore. However, some of them remarked that although the farming techniques had largely changed, they kept in mind lessons of the past: "*You always remember what ancestors told you and you follow such advice. It's good to know about everything: the new and the old techniques*". Moreover, those cultivating cereals (normally in large plots) and using mineral fertilizers were aware of their management's negative impacts on the

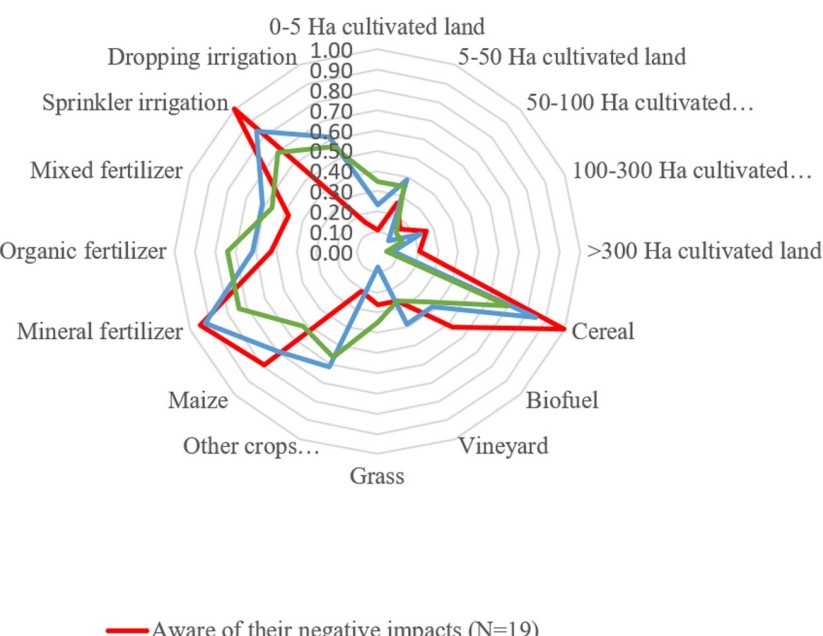

**Fig 3. Land management differences among the groups of farmers with different NCP co-production awareness.**

agrarian ecosystem. Interviews showed that those aware of their negatives effects particularly thought they were eroding and polluting the land.

Most modern technology adopters agreed that the new irrigation was favoring the appearance of new pests and they needed to use higher doses of pesticides, despite being aware of its negative effects on habitats and biodiversity. One of the modern irrigation adopters explained "*I understand that every year, 2–3 of the products we use are prohibited because they are bad for our health and the environment. However, there is no real support to shift to organic farming at a large-scale, which is necessary because if your neighbors apply those products, they make it really difficult for you. That is, if I started organic farming, all the slugs would come to my plot. The slugs will eat all my maize. I am now using transgenic maize to prevent this. Moreover, when selling my crops to the Germans, if they find other (illegal) products in the organic farming requisites, not only will they not buy my crop but I will also get a sanction. If organic farming is not supported at large-scale I will not risk my money. Nowadays, there is only propaganda but no real support*".

Some of the traditional farmers also admitted they needed to use conventional pesticides because of the amount of pests in their plots. One of them stated: "*I believed that the herbicides and fungicides we use include a component to make the pests resistant so that we need to increase the doses we apply. I have a plum tree that when my father was alive we didn't apply anything and we got extraordinary plums; nowadays, if I don't apply products, I don't get anything*".

Few traditional and modern farmers elaborated their ideas regarding how they affected land erosion. From those, all traditional farmers explained they benefitted the control of erosion through their management: "*I take care of the crops by making rotations to favour the retention of the soil*". Many modern farmers (55%) thought they were eroding their lands through the labours and the type of irrigation they did, whereas only some (33%) thought their management positively affected the regulation of erosion: "*If there are crops, there is no erosion. I grab the soil through the crops roots I grow*".

Regarding the capacity of land to regulate land and underground water pollution, few traditional farmers and landholders explained their perceptions in this regard, whereas some modern farmers had contrasting viewpoints: some thought they benefitted the agrarian ecosystem because of the type of crops they selected, the rotations they conducted and the amount of vegetal mass they had in their land: "*I grow regulating species to get the CAP payment*", whereas others thought they were polluting the agroecosystem.

When assessing how different types of farmers (in terms of MC) perceived their NCP co-production, we neither found any significant difference among the group of farmers but we did find similar tendencies complemented by qualitative insights: traditional farmers (who cultivate small plots, grow vegetables and fruit trees with drip irrigation systems and mixed fertilizers) had a higher awareness of their role in enhancing traditional knowledge, traditions, and landscape aesthetics (see Fig 4). Likewise, (but to a lower extent) they also thought they contributed to climate regulation. For more details on those differences see S3 and S4 Tables.

Both, traditional and modern farmers thought they contributed to landscape aesthetics. As previously commented, most traditional farmers related planting trees with the improvement of landscape. Modern farmers linked this contribution with the adoption of large-scale irrigation: "*Before (Itoiz-Canal de Navarra) you arrived to this village in summer and it looked like a desert. We used to finish harvesting in San Fermin and everything was yellow, none can imagine how much those villages have changed. Nowadays, you wake up in the morning and see all the fields irrigated, the maize so green*".

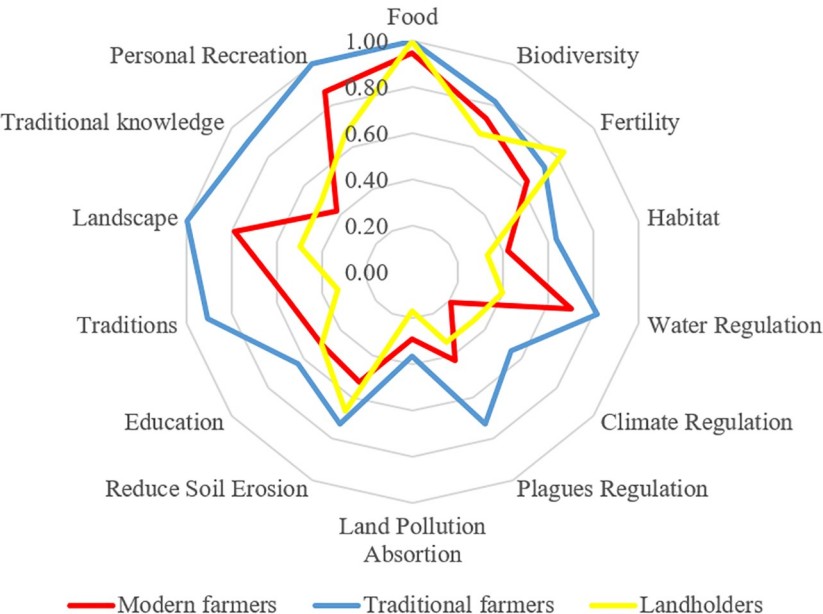

**Fig 4. NCP co-production awareness differences among the groups of farmers with distinct land management strategies.**

## Exponential Random Graph Model (ERGM) social network results

The results of the ERGM (Table 2) indicate that knowledge on farming practices seeking is strongly reciprocal; that is, farmers are likely to ask advice from farmers that also ask them for advice in return, and highly transitive, as can be seen in the positive gwesp term. Neither gwin- nor outdegree are significant, that is, there are no significant hubs in the network.

Farmers sharing the same crops are more likely to seek knowledge on farming practices from each other while farming land close to each other or the age-difference does not significantly influence the knowledge-seeking behavior.

The results for the MC reveal that old landholders (MC3) are more likely to be asked for farming practices knowledge (reference category MC1), and traditional farmers (MC2) and old landholders (MC3) are less likely to ask for such a knowledge. For the NCPC awareness clusters, we find that farmers who think they contribute to climate regulation and landscape aesthetics (NCPC2) are less likely to be asked for their knowledge on farming practices, and more likely to ask others for such a knowledge. There seems to be no preference to ask farmers within the same NCPC or management behavior for advice.

Additionally, a more detailed analysis of the knowledge-seeking behavior modeling the connectivity between the management clusters revealed that traditional farmers (MC2) in this sample do not ask each other for knowledge on farming practices, nor do they ask old landholders (MC3). Likewise, old landholders (MC3) do not seek knowledge from traditional farmers (MC2). These results indicate that modern farmers (MC1) are crucial for the information flow throughout the community.

Complementary to ERGM results, Fig 5 reveals that modern farmers (in red) occupy central positions and are very active in both giving and receiving knowledge about land management and are often aware of their negative effect on the agrarian ecosystem. In contrast, most traditional farmers are in the periphery and often isolated, not being asked or asking for knowledge to others. Such traditional farmers (in blue) are often aware of their role to support the

**Table 2. ERGM results for farmers knowledge-seeking presenting the estimated value in the model for each of the parameters considered (Estimate), their standard error (SE) and the p-value (p).**

| Parameter | Estimate | SE | *p* |
|---|---|---|---|
| **Edges (ties)** | **-5.594** | **0.673** | **<.001** |
| **Mutual (reciprocity)** | **7.472** | **0.638** | **<.001** |
| **gwesp($\alpha$ = 0.3) (transitivity)** | **0.457** | **0.121** | **<.001** |
| gw-indegree ($\alpha$ = 0.9) (popularity) | 0.098 | 0.505 | .845 |
| gw-outdegree ($\alpha$ = 0.1) (activity) | -0.770 | 0.544 | .157 |
| absolute age difference (homophily age) | -0.011 | 0.006 | .083 |
| **shared crops** (homophily crops) | **0.207** | **0.076** | **.006** |
| distance (1km) | -0.147 | 0.143 | .304 |
| indegree MC2[a] | 0.559 | 0.693 | .420 |
| **indegree MC3**[a] | **1.363** | **0.672** | **.043** |
| **outdegree MC2**[a] | **-2.463** | **0.753** | **<.001** |
| **outdegree MC3**[a] | **-2.637** | **0.671** | **<.001** |
| same MC (homophily management) | -0.167 | 0.488 | .732 |
| **indegree NCPC2**[b] | **-0.897** | **0.366** | **.014** |
| indegree NCPC3[b] | -0.114 | 0.411 | .781 |
| **outdegree NCPC2**[b] | **0.942** | **0.351** | **.007** |
| outdegree NCPC3[b] | 0.063 | 0.425 | .883 |
| same NCPC (homophily perception) | -0.190 | 0.140 | .175 |

Note. AIC: 784.5 BIC: 906.

[a]Reference category: MC1.

[b]Reference category: NCP1.

agrarian ecosystem habitat, as well as those aware of their capacity to regulate climate and contribute to landscape configuration (Figs 3–5). Therefore, we can deduce that most farmers aware of their regulating and cultural NCP do not play an important role in this network. However, modern farmers control the information flow, being able to spread intensive farming practices in this community. Fig 5 allows us to visualize how farmers performing intensive practices dominate over traditional practices, even when some of such modern farmers recognize their impacts on the agrarian ecosystem.

## Discussion and conclusions

We found that most farmers were aware of their co-production of NCP through their land management decisions, though modern and traditional farmers' awareness of their contributions differed. We found a higher degree of homogeneity among traditional farmers in terms of their perceptions about non-material NCP, such as climate change regulation, traditional knowledge, cultural traditions, and landscape aesthetics contribution. In addition, we found much greater heterogeneity in terms of the awareness about NCP among modern farmers cultivating large plots (>100 Ha) especially regarding the negative ecological and cultural impacts of their management on non-food NCP. Further, while many modern farmers were not aware of their role in co-producing non-food NCP, and some (around 30–40% of those answering) recognised their negative effects, still others justified their management based on their perceived overall positive impact in the agrarian ecosystem.

We also found that farmers' awareness about NCP co-production and their land management decisions were correlated with the structure of the social networks among the farming

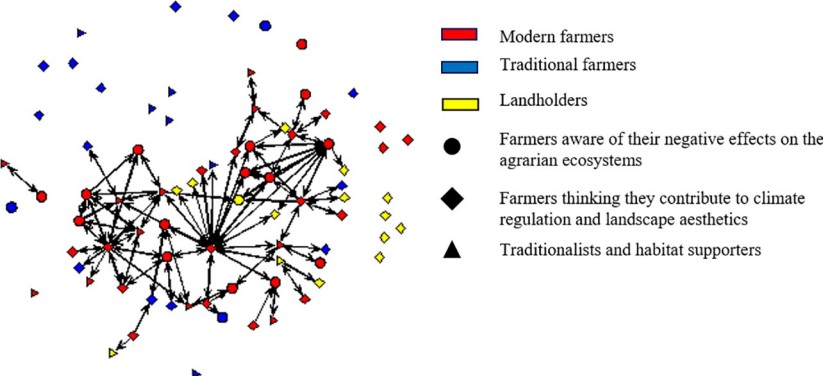

**Fig 5. The knowledge on farming practices network of the farming community in the village of Navarre in 2017.**

community. Farmers sharing similar land management practices and awareness of NCP co-production affect ties within the social network. Intensive farmers were most active and took central positions in the exchange of farming knowledge. In contrast, traditional farmers were often more isolated within the network (Fig 5), not sharing their farming knowledge with the rest of the farmers. This implies that those farmers who were more conscious about the social-ecological implications of their land management were *less likely to be sought for knowledge on farming practices* by other farmers and thus less prone to disseminate their knowledge and perceptions among the community of farmers, as compared to the modern farmers located more centrally in the social network.

These networks are used to promote climate mitigation [24] and could be used in a similar way for the spread of (un)sustainable farming practices. Our results point towards the roles and responsibilities of modern vs. traditional farmers when both kinds of farmers choosing different co-production pathways [53]. Moreover, the cognitive and management dimensions linked to knowledge sharing are critical to understanding how dominant farming practices (intensive practices in this case), can be spread at the landscape scale [24, 25].

Several factors could help explain the social mechanisms behind our results. In line with what other authors, such as [52] found in the Philippines, despite higher investment and labor requirements, modern farmers find large-scale irrigation attractive, losing the longer term perspective of the benefits associated with traditional and diversified farming despite educational programs. This result aligns with what [34] found and which points towards the idea that it is not always a lack of knowledge that prevents changes in agricultural practices. In other words, just sharing knowledge with dominant farmers is necessary but not sufficient. What is needed for understanding the spread of intensive practices is a focusing on the more systematic and structural causes that make dominant intensive farming to spread [53]. The integration of rural economies into global commodity markets makes farmers increasingly dependent on enhancing yields through intensive practices [1, 54, 55]. It is at the national and international incentive structures that promote agricultural commoditization and agribusiness models that require increased scrutiny and change to favour farmers' awareness and the distribution of their influence in spreading farming practices at the landscape level. On the positive side, we also found that some modern farmers would be willing to shift the way they use agrochemicals if there were real institutional support for such change aligned with their larger scale operations. This finding reveals that as [53] found, one of the main drivers for land management decisions at local scale is still government intervention to provide clear economic incentives and regulation.

The introduction of large-scale irrigation seems to be exposing those traditional farmers to unexpected losses in local knowledge and the substitution of crops not suiting carbon store capacity, like fruit trees [11]. Further research is needed regarding how those types of farms are becoming invisible for policies and markets, making at the same time very difficult their interests be supported, and consequently, their intangible contributions, such as traditional knowledge, can be lost over time [56].

Reciprocity and transitive closure playing an important role in the network are typical findings (Siciliano, 2015; Thomas and Caillon, 2016). Giving knowledge and providing help costs resources and is often repaid by knowledge and help in return. Modern farmers controlled the flow of information since traditional farmers and landholders did not normally ask each other for advice. Modern farmers sought, however, more advice from landowners than they did from each other or traditional farmers. This finding may be related to the fact that modern farmers normally cultivated landholders' plots and, therefore, asked them about their land management preferences.

In contrast, exogenous factors, such as proximity to other farmers, were not a driving factor in forming network structure (as happened [57] in their case study about agroforestry advice networks in Ghana). Houses proximity may be more important than the proximity of the cultivated plots or easily available modern technology (e.g., mobile telephones) reduces the importance of shorter distances.

It can be difficult to make generalizations with a case study of a small community. However, we find important contextual similarities with other rural communities in Western Europe [56] and worldwide [54]. Additionally, we did not have longitudinal data on the networks and attributes. This reduce our ability to assess how advice networks in rural communities over time may influence farmers' land management decisions and how dynamic variables, including changes in awareness about climate change, may affect the formation of rural networks (see, e.g., [58], for the problem of how to disentangle network selection and network influence processes).

Despite these caveats, rural network analysis can be useful for understanding the network configuration of rural farming communities to improve rural policy development since it permits understanding interactions between awareness, land management decisions, and knowledge/advice sharing at the landscape level. Being aware of modern farmers' position that enables them to control the flow of knowledge shows how difficult it can be to spread less intense farming practices [25]. Considering that those modern farmers' perceptions and their management practices are significant factors for the creation of the advice network structure, we should incentive structures that make farmers more aware of their contribution to climate regulation to take a more active role within their networks. Incentivizing traditional farmers' practices diffusion through their higher participation in advice-giving structures (e.g. within the local cooperatives), as well as withdrawing some power from modern farmers could be one way of changing this situation.

## Supporting information

**S1 File. Farmers, landholders and rural organizations interviews.**
(DOCX)

**S1 Table. Characterization of one of the clusters regarding farmers' land use management.**
(DOCX)

**S2 Table. Environmental variables considered being included in the model.**
(DOCX)

**S3 Table. Land management differences among the groups of farmers with NCP co-production awareness.**
(DOCX)

**S4 Table. ERGM to check homophily in the type of farming effect in ties formation.**
(DOCX)

**S5 Table. ERGM to check to belong to uneven types of farming effect in ties formation.**
(DOCX)

**S1 Dataset. Original dataset and main analysis.**
(XLSX)

# Acknowledgments

A.A. wishes to express her gratitude to her mother, Gloria Aguinaco Otxaran, who has taken care of her first baby while she working on this project. We would also like to thank two anonymous reviewers and the editor for helpful comments and advice to improve the paper.

# Author Contributions

**Conceptualization:** Amaia Albizua, Unai Pascual.

**Data curation:** Amaia Albizua, Guillaume Larocque.

**Formal analysis:** Amaia Albizua, Guillaume Larocque, Robert W. Krause.

**Funding acquisition:** Amaia Albizua, Elena M. Bennett.

**Investigation:** Amaia Albizua, Elena M. Bennett.

**Methodology:** Amaia Albizua.

**Project administration:** Amaia Albizua, Elena M. Bennett.

**Supervision:** Elena M. Bennett, Unai Pascual.

**Validation:** Amaia Albizua, Unai Pascual.

**Visualization:** Amaia Albizua.

**Writing – original draft:** Amaia Albizua.

**Writing – review & editing:** Amaia Albizua, Elena M. Bennett, Guillaume Larocque, Robert W. Krause, Unai Pascual.

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
