## [Decision Letter · Decision Letter 0]

10 Sep 2020

PONE-D-20-24320

What affects farmers’ advice networks: implications for agrarian sustainability

PLOS ONE

Dear Dr. Albizua,

Thank you for submitting your manuscript to PLOS ONE. After careful consideration, we feel that it has merit but does not fully meet PLOS ONE’s publication criteria as it currently stands. Therefore, we invite you to submit a revised version of the manuscript that addresses the points raised during the review process.

As noted in the editorial comments, there is a lack of detail regarding the statistical analyses undertaken, and the qualitative data collected is almost entirely missing in the results. These are two very important aspects that require attention. There are additional helpful comments from the reviewers which need to be addressed.

We look forward to receiving your revised manuscript.

Kind regards,

Sieglinde S. Snapp

Academic Editor

PLOS ONE

Journal Requirements:

'The case study had the required university ethics approvals from McGill University.'

a. Please amend your current ethics statement to include the full name of the ethics committee/institutional review board(s) that approved your specific study and confirm that your named institutional review board or ethics committee specifically approved this study.

b. Once you have amended this statement in the Methods section of the manuscript, please add the same text to the “Ethics Statement” field of the submission form (via “Edit Submission”).

4. Please ensure that you refer to Figure 1 in your text as, if accepted, production will need this reference to link the reader to the figure.

Additional Editor Comments (if provided):

The reviewers were split in their assessment, reviewer one recommended reject and reviewer 2 recommended minor revision. I reviewed and agree in the main with reviewer one, that the statistics section is not written with sufficient detail so it is quite difficult to discern how the study was conducted. Also, that the study would have benefited from qualitative data, which was collected based on the methods description, yet not reported on. So I recommend a major revision addressing these points, and indeed all the reviewers comments - at which time it will be possible to ascertain the statistical validity of the study and the key insights derived from it.

Reviewers' comments:

Reviewer's Responses to Questions

**Comments to the Author**

1. Is the manuscript technically sound, and do the data support the conclusions?

Reviewer #1: Partly

Reviewer #2: Yes

2. Has the statistical analysis been performed appropriately and rigorously? 

Reviewer #1: No

Reviewer #2: Yes

3. Have the authors made all data underlying the findings in their manuscript fully available?

Reviewer #1: No

Reviewer #2: Yes

4. Is the manuscript presented in an intelligible fashion and written in standard English?

Reviewer #1: Yes

Reviewer #2: Yes

5. Review Comments to the Author

Reviewer #1: Summary of manuscript

- In this study, the authors investigate a farming community in Spain undergoing agricultural intensification to identify farmers’ perceptions of their impacts on the local ecosystem and how farmer social network ties differ across different management groups. Overall this study addresses important implications around farmer social networks and how these dynamics may contribute to dominant management practices. However, the connections made by the authors are founded in weak statistical associations, and the major conclusions claimed by this study overstate these weak associations. Specifically, the main results presented highlight differences in management practices and NCP awareness for different groups of farmers, however according to the authors these differences did not hold up after accounting for multiple comparisons error, therefore they cannot be presented as differences as the main conclusions state. For this reason, I do not recommend publication for this manuscript.

Abstract and Introduction

- Abstract does summarize research questions and key findings, however much of the manuscript focuses on Nature Contributions to People (NCP) concept, and this is not mentioned in the abstract.

- Overall language editing is recommended to improve clarity and sentence structure.

- The introduction should include more literature focusing on the linkages between agricultural intensification practices being both unsustainable/environmentally degrading and the authors’ argument that these stronger social networks undermines/pushes out small-scale organic farmers who could be creating more positive environmental interactions in the landscape.

- Introduction needs more synthesis of literature – expand on previous studies that look at topics mentioned. Additionally, the authors have not properly formatted the citations, such that numbered in-text citations do not have matching numbered references in references section. This makes it difficult to assess the literature being cited and identify key references.

- Current structure includes Introduction and Conceptual Framework. For the structure of this journal, include conceptual framework as part of introduction, not a separate section.

Figures and Tables

- Figures 2 and 3 do not appear to be necessary – just show sampling of participants for social network analysis and sampling of interview participants. This can just be described in the methods.

- Table 1 – Characteristics of groups determined by HCA - How were characteristics compiled? Thematic analysis of interviews?Interview response analysis not described in methods

- Figures 5 & 6 – Contain confusing captions that do not match with description of radar charts. Captions state that results of statistical differences found before accounting for multiple comparisons, given differences found did not hold up after accounting for multiple comparisons error, authors should not indicate statistical differences.

- Figure 7 – Advice network of the farming community: Unclear figure – caption does not accurately describe figure – more description needed to understand figure. Does not clearly demonstrate results

- Supplementary material not appropriately labeled given journal conventions.

Methods

- Line 186 - Semi-structured interviews - Unclear how the results of these interviews were used – This data does not appear to be represented in results section?

- Line 197 – Hierarchical Cluster Analysis - More detail on method needed – variables used, number of farms subjected to analysis, method used to select groups etc.

- Overall there is not enough detail in methods description to determine if statistical analysis has been performed rigorously. More detail is needed to explain statistical approach.

Results, discussion, conclusion

- In general, tables and figures should be better supported in the text. There is not enough description of results in text.

- More literature needs to be connected to discussion. What have other studies looking at agricultural intensification found as far as dynamics between large-scale intensified farms and those maintaining traditional practices?

- Conclusion seems to be an over-reach given the results and weak statistical basis.

- References - Improperly formatted – Numbers not listed in reference section, instead references organized by author, therefore unable to connect citations in text to reference list. This is a major oversight that makes critical examination of the literature difficult.

Overall, the subject of this study is valid and the authors have sufficient data to analyze, however I think this study could benefit from a more mixed-methods approach, incorporating more qualitative data from interviews collected to complement the quantitative approach used here.

Reviewer #2: General comment

Relevant paper, well-written. Methods appropriate but analysis could dig deeper. The data collected are new and appear to be well collected and analyzed. The material is quite informative and does indeed significantly add to our knowledge on the role of social networks. These findings are well discussed in the paper, with some surprising and quite interesting findings emerge, including role of social networks, farmer perceptions and awareness for the distinct groups of farmers. The tables and figures on sample selection and characteristics are informative.

Some more specific comments

study area the authors should:

• Provide more general characteristics of the region. Since the village identity has to remain hidden.

• Line 142 and 143: explain the increase in land from what was used previously and what is the current expansion?

• Line 144- 145: expand by explaining the crops grown previously that lead to the new crops adopted. the authors should explain the rate of fertilizer that was used previously and the new rate.

• Line 146 to 149: explain what is the rate of nitrate here?

Data and methods

the authors should:

• Line 166-167: explain the reasons of receiving no response for the group of farmers that was initially selected and lead to the selecting more farmers.

• Explain how the researchers controlled for bias in the selection of the 80 additional people.

• Line 168-170: explain the criteria’s used in nominating the additional people in the survey.

• Line 174: show the number of interviews done for each group of farmers and landholders?

• Line 230: did the authors check for robustness after removal of the insignificant variables?

Results

• The opening line, 284 talks about awareness, but the heading is perception this contradicts with the next section that discusses awareness?

• Line 286-291: mention the percentage for the highest and lowest levels in the results shown in the figure.

• Line 296 is contracting, were they two or three distinct groups?

• Line 295-309: The figures in the parenthesis for intensive, small-scale farmers and landowners do not match with those provided in table 1 (HCA1 based on management)

conclusion and discussion

The authors doesn't explain in the discussion section how the results link to agriculture intensification, more information is needed..

6. PLOS authors have the option to publish the peer review history of their article (what does this mean?). If published, this will include your full peer review and any attached files.

Reviewer #1: **Yes: **Alison Nord

Reviewer #2: No

---

## [Author Response · Author response to Decision Letter 0]

29 Oct 2020

We are grateful to the editors and the two reviewers of our manuscript for their insightful and useful comments. We provide below a detailed account of how we have addressed each of their remarks. We think that the manuscript now better explains the statistical analyses undertaken, and we have added more insights about the qualitative data collected. We hope the reviewers find the resubmitted manuscript stronger and sharper. We have especially worked to link the introduction and discussion to the literature connecting agricultural intensification practices with community social networks to show how intensification may push out more traditional, small-scale farmers. We hope the reviewers agree that the concluding discussion is now more robust in acknowledging that while some of our findings are not statistically significant, the qualitative research points at some trends that are worth noting. We hope that the re-submitted manuscript meets now the required scientific standards of PlosONE. Please do not hesitate to contact us for any further clarification on how we have interpreted or addressed the comments below.

Yours sincerely,

Dr. Amaia Albizua

Dr. Elena Bennett

Dr. Guillaume Larocque

Dr. Robert Krause

Dr. Unai Pascual

Amended, style and file naming has been corrected.

2. Thank you for including your ethics statement: 'The case study had the required university ethics approvals from McGill University.'

a. Please amend your current ethics statement to include the full name of the ethics committee/institutional review board(s) that approved your specific study and confirm that your named institutional review board or ethics committee specifically approved this study. 

We included in Methods section: “The McGill University Faculty of Agriculture and Environmental Sciences Research Ethics Board reviewed and approved this project by delegated review in accordance with the requirements of the McGill University Policy on the Ethical Conduct of Research Involving Human Participants and the Tri-Council Policy Statement: Ethical Conduct For Research Involving Humans.”

b. Once you have amended this statement in the Methods section of the manuscript, please add the same text to the “Ethics Statement” field of the submission form (via “Edit Submission”).

Amended.

We have given a doi to the original data and the analysis performed. This is S7 in the supplementary material. https://doi.org/10.6084/m9.figshare.13089098.v1

4. Please ensure that you refer to Figure 1 in your text as, if accepted, production will need this reference to link the reader to the figure. 

Amended.

Amended.

Additional Editor Comments (if provided):

The reviewers were split in their assessment, reviewer one recommended reject and reviewer 2 recommended minor revision. I reviewed and agree in the main with reviewer one, that the statistics section is not written with sufficient detail so it is quite difficult to discern how the study was conducted. Also, that the study would have benefited from qualitative data, which was collected based on the methods description, yet not reported on. So I recommend a major revision addressing these points, and indeed all the reviewers comments - at which time it will be possible to ascertain the statistical validity of the study and the key insights derived from it.

We have added more detail in the methods section, regarding how the statistics were analyzed and the study conducted. We now explain that there were two kinds of interviews for data gathered in the research. One conducted among farmers and landholders and the other one conducted among rural organizations. Only the data coming from farmers and landholders was used to make the social network analysis and the statistics about farmers’ awareness regarding their role in co-producing NCP (see page 10 and 11 in the Manuscript). We have now added a section called: “Semi-structured interviews response analysis” (page 10 and 11 in the Manuscript). Here we explain how we used information from the surveys and how such data feeds a qualitative analysis to complement the results obtained through statistics. Please see below more detailed responses to the reviewers.

Reviewer 1

- In this study, the authors investigate a farming community in Spain undergoing agricultural intensification to identify farmers’ perceptions of their impacts on the local ecosystem and how farmer social network ties differ across different management groups. Overall this study addresses important implications around farmer social networks and how these dynamics may contribute to dominant management practices. However, the connections made by the authors are founded in weak statistical associations, and the major conclusions claimed by this study overstate these weak associations. Specifically, the main results presented highlight differences in management practices and NCP awareness for different groups of farmers, however according to the authors these differences did not hold up after accounting for multiple comparisons error, therefore they cannot be presented as differences as the main conclusions state. For this reason, I do not recommend publication for this manuscript.

Thank you for sharing this important comment. We would like to clarify that while (only) the results about farmers’ perceptions are not statistically significant, we approach farmers’ perceptions qualitatively, making this point more clear. We find out that the type of management and farmers’ awareness are significant variables when included in the Exponential Random Graph Model (ERGM). Thus, we believe that it is pertinent to highlight the results regarding what matters for the creation of community networks and how some types of farmers have a dominant role in the sharing of information, which we believe is grounded on data.

Abstract and Introduction

- Abstract does summarize research questions and key findings, however much of the manuscript focuses on Nature Contributions to People (NCP) concept, and this is not mentioned in the abstract.

Well spotted and amended. We have changed the sentence where we said that few studies address farmers’ perceptions about their influence on agrarian ecosystems by: farmers’ awareness regarding their role to co-produce NCP.

- Overall language editing is recommended to improve clarity and sentence structure.

Thank you. We have asked a native speaker to help us with the editing of the manuscript.

- The introduction should include more literature focusing on the linkages between agricultural intensification practices being both unsustainable/environmentally degrading and the authors’ argument that these stronger social networks undermines/pushes out small-scale organic farmers who could be creating more positive environmental interactions in the landscape. Introduction needs more synthesis of literature – expand on previous studies that look at topics mentioned.

Thank you for the suggestion. We have included references to the literature including the works by Villanueva et al. 2017; Isaac 2017; Ernstson et al. 2010; Isaac and Matous 2017 and Albizua et al 2020.

Those studies show examples about how farmers’ networks influence their management which simultaneously is connected to land change and biodiversity changes. They correlate such environmental changes with the number and characteristics of the network ties. One example also talks about how multi-level network structures are correlated with the access to information and resources for different kinds of land management being able to push out more vulnerable rural livelihoods.

Additionally, the authors have not properly formatted the citations, such that numbered in-text citations do not have matching numbered references in references section. This makes it difficult to assess the literature being cited and identify key references.

Amended. We are sorry for this mistake, we forgot to refresh the bibliography and left an old version of it in the manuscript. This is amended now.

- Current structure includes Introduction and Conceptual Framework. For the structure of this journal, include conceptual framework as part of introduction, not a separate section.

Thank you. We agree and we have now merged the two to follow journal guidelines on structure.

Figures and Tables

- Figures 2 and 3 do not appear to be necessary – just show sampling of participants for social network analysis and sampling of interview participants. This can just be described in the methods.

- Table 1 – Characteristics of groups determined by HCA - How were characteristics compiled? Thematic analysis of interviews? Interview response analysis not described in methods

- Figures 5 & 6 – Contain confusing captions that do not match with description of radar charts. Captions state that results of statistical differences found before accounting for multiple comparisons, given differences found did not hold up after accounting for multiple comparisons error, authors should not indicate statistical differences.

- Figure 7 – Advice network of the farming community: Unclear figure – caption does not accurately describe figure – more description needed to understand figure. Does not clearly demonstrate results

Thank you for the suggestions. We have taken out figures 2 and 3 and kept the main points about sampling in the methods section. We have now added information in the methods section about how data collection (via surveys) about the different characteristics about land management (page 9). We have now included information about how we conducted the interviews response analysis (a new sub-section added). As for Figures 5 and 6 (now Figures 3 and 4), we follow the recommendations and have corrected the captions. Lastly, also following the reviewer’s suggestion we have improved the caption of the old figure 7 (now Figure 5) and added an additional description in the main text to better explain the information that can be obtained from this figure.

- Supplementary material not appropriately labeled given journal conventions.

Thank you. We have now amended the way the Supp. Material is labelled and added captions with online hiperlinks.

Results, discussion, conclusion

- In general, tables and figures should be better supported in the text. There is not enough description of results in text. 

We have revised the text thoroughly and tried to connect the information from tables and figures with the main text (see track of changes document).

Methods

- Line 186 - Semi-structured interviews - Unclear how the results of these interviews were used – This data does not appear to be represented in results section?

- Line 197 – Hierarchical Cluster Analysis - More detail on method needed – variables used, number of farms subjected to analysis, method used to select groups etc.

- Overall there is not enough detail in methods description to determine if statistical analysis has been performed rigorously. More detail is needed to explain statistical approach.

Thank you for these questions. The semi-structured interviews provided the data for the network analysis and responses were transcribed and analyzed taking an inductive approach. 

We have now added a section called “interview qualitative analysis”, explaining how the qualitative analysis was done in the methods section and the results have been added at the end of the section called “Differences among farmer groups regarding management and NCP co-production awareness” (pages18,19 and 20 in the manuscript) 

The qualitative information shows how traditional farmers seem to be more connected to landscape aesthetics and climate regulation due to the kind of farming they conduct which offers different benefits at the landscape level being more sustainable than other farming paths chosen by other kind of farmers.

We have now included information about the hierarchical cluster analysis used as requested. Moreover, we give an example in the supplementary material about how we have interpreted the Hierarchical cluster analysis results.

Given the above, we hope that the statistical analysis explained in a more clear way within the methods section.

- More literature needs to be connected to discussion. What have other studies looking at agricultural intensification found as far as dynamics between large-scale intensified farms and those maintaining traditional practices?

Thank you for the suggestion to link the discussion from our results to the broader literature touching on this topic. We have now included additional literature, e.g., Kay, 2002 who point towards the roles and responsibilities choosing different co-production pathways.

Martinez-Baron et al., 2018 regarding how climate mitigation practices can spread and scale up, or Albizua et al., 2020 regarding how some intensive farmers displace more traditional and small-scale farmers due to the intensification taking place in the region. 

We have also included Chen at al, 2018; Kay, 2002; Cramb, 2011; to explain more systematic and structural causes that make dominant intensive farming to spread at the cost of marginalizing other farming strategies.

- Conclusion seems to be an over-reach given the results and weak statistical basis.

Thank you. We hope the new conclusion section is now more accurate and humble with regard to the intend of the paper and the way the results connect with the main research question. 

- References - Improperly formatted – Numbers not listed in reference section, instead references organized by author, therefore unable to connect citations in text to reference list. This is a major oversight that makes critical examination of the literature difficult.

We are sorry for this mistake. This oversight was due to using a different citation style at the beginning which was not corrected at the time of submission. We apologize. We have now made sure to provide the right formatting to the reference list and citations throughout the paper. 

We have made a final review in the manuscript and corrected some few references that appear more than once wrongly cited. This last correction only appears in the final manuscript version but not in the track of changes. The manuscript has changed so much that revising all the references twice will take us a lot of time and it is already done in the manuscript without track of changes. We hope this is not a problem.

Overall, the subject of this study is valid and the authors have sufficient data to analyze, however I think this study could benefit from a more mixed-methods approach, incorporating more qualitative data from interviews collected to complement the quantitative approach used here.

Thank you for the critical yet very constructive comments of the reviewer. We have also incorporated some more qualitative data from the surveys to complement the quantitative approach, as well as to shed some further light on those statistical results that while not statistically significant, are still interesting to focus on from a qualitative perspective. We believe those qualitative insights can offer complementarity perspective on the stat results.

Reviewer 2

Reviewer #2: General comment

Relevant paper, well-written. Methods appropriate but analysis could dig deeper. The data collected are new and appear to be well collected and analyzed. The material is quite informative and does indeed significantly add to our knowledge on the role of social networks. These findings are well discussed in the paper, with some surprising and quite interesting findings emerge, including role of social networks, farmer perceptions and awareness for the distinct groups of farmers. The tables and figures on sample selection and characteristics are informative.

Thank you for this overall comment on the paper. We have now gone deeper into the analysis, also requested by Reviewer 1, adding some qualitative information from the surveys (see methods and results section) and we have tried to more clearly connect the results from the analysis with literature linking agricultural intensification and dynamics between large-scale intensified farmers and those maintaining more traditional and sustainable agricultural management practices.

Study area the authors should:

• Provide more general characteristics of the region. Since the village identity has to remain hidden.

• Line 142 and 143: explain the increase in land from what was used previously and what is the current expansion?

• Line 144- 145: expand by explaining the crops grown previously that lead to the new crops adopted. the authors should explain the rate of fertilizer that was used previously and the new rate.

• Line 146 to 149: explain what is the rate of nitrate here?

We have provided more general information about the location including e.g, the climatic conditions, the large-scale irrigation project and the total area converted into modern irrigation as well as the impact on the available arable land area due to this development project, as suggested.

We now explain that traditional farmers used to grow vegetables and fruit trees such as olive and almond trees. However, fertilizers rates vary a lot depending on the farmer and we unfortunately failed to collect this information. We know that fertilizers doses increased since most of farmers and the rural advisor informed about this during the interviews but they did not provide exact figures.

Several authors studying nitrate pollution in the region mention that agricultural intensification is leading to higher nitrate pollution of land and water (see e.g., (Ladrera et al., 2019)).

50 mg/ l is the threshold to say the water is polluted by nitrates attending to the Spanish Royal Decree 261/1996, 16 Feb, (for more information see article #3 in https://www.mapa.gob.es/es/agricultura/legislacion/RD_261_1996_tcm30-73046.pdf) 

Data and methods the authors should:

• Line 166-167: explain the reasons of receiving no response for the group of farmers that was initially selected and lead to the selecting more farmers.

Thank you for asking to provide more information on this issue. We got a response rate of 77% through the combination of a priori-list and the follow up snowball sampling (see page 9 L172 in the manuscript). Additional people was mentioned by the contacted farmers when we collected information for the social network analysis (N=161) but we just conducted the ERGM analysis on the data gathered from those farmers we initially approached (n=81 farmers). Since NCP is the main focus of this study, we decided to only rely on the data from the subsample (n=81), which we believe is enough data from the case study to provide reliable results (see e.g., Krause et al. 2020 in Social Networks)

• Line 168-170: explain the criteria’s used in nominating the additional people in the survey.

Thank you for the suggestion. We used a free recall system. This is, when we asked to mention up to five other people who they think influence their land management decision (see page 10 L190 in the manuscript). They were not provided a list of names.

• Line 174: show the number of interviews done for each group of farmers and landholders?

We have now clarified that we made the 81 interviews (to all farmers and landholders who wanted to respond and then we classified them) (see page 9 L172 L344 in the manuscript). The numbers for each group appeared in the Results section, in the subsection called “Differentiated groups of farmers”, there you can see: modern farmers (N=45), traditional farmers (N=21), and landowners (N=15) (see table 1 in page 16. L344 in the manuscript)

• Line 230: did the authors check for robustness after removal of the insignificant variables?

We present the results of the full model, including the two previously removed variables, in table 2. The results are robust to inclusion/exclusion of these variables.

• Explain how the researchers controlled for bias in the selection of the 80 additional people.

We tried to approach all the farmers and landholders in the community, trying to cover the whole range of diversity in terms of land management strategies. They were free to mention anyone they considered influential for their land management.

Results

• The opening line, 284 talks about awareness, but the heading is perception this contradicts with the next section that discusses awareness?

We have corrected the language so as to not confuse the readers. 

• Line 286-291: mention the percentage for the highest and lowest levels in the results shown in the figure.

Done. 

• Line 296 is contracting, were they two or three distinct groups?

You are right, it is confusing. Each HCA found 3 distinct groups of farmers. Corrected now.

• Line 295-309: The figures in the parenthesis for intensive, small-scale farmers and landowners do not match with those provided in table 1 (HCA1 based on management)

Well spotted, thank you. This information is now corrected. 

conclusion and discussion

The authors doesn't explain in the discussion section how the results link to agriculture intensification, more information is needed.

Thank you. We have done this now.

We explain that modern farmers, who normally perform more intense farming techniques, were the most active and central in the network to exchange farming knowledge enabling them to spread the “know-how” of their farming strategies whereas traditional farmers who are more isolated don’t share their farming knowledge with the rest of the farmers. This implies that those farmers who were more conscious about the social-ecological implications of their land management were less likely to be sought for knowledge on farming practices by other farmers and thus less prone to disseminate their knowledge and perceptions among the community (see page 24 L479:486). Moreover, we have now included some new references such as: Kay, 2002 who point towards the roles and responsibilities choosing different co-production pathways, or Martinez-Baron et al., 2018 regarding how climate mitigation practices or Albizua et al., 2020 regarding how farming techniques can spread and scale up.

We have also included Chen at al, 2018; Kay, 2002; Cramb, 2011; to explain more systematic and structural causes that make dominant intensive farming to spread.

---

## [Decision Letter · Decision Letter 1]

14 Dec 2020

Social networks influence farming practices and agrarian sustainability

PONE-D-20-24320R1

Dear Dr. Albizua,

We’re pleased to inform you that your manuscript has been judged scientifically suitable for publication and will be formally accepted for publication once it meets all outstanding technical requirements.

Kind regards,

Sieglinde S. Snapp

Academic Editor

PLOS ONE

Additional Editor Comments: I agree with the reviewers that the revised manuscript is greatly improved, we appreciate the careful attention to making these changes and addressing all the comments.

Reviewers' comments:

Reviewer's Responses to Questions

**Comments to the Author**

1. If the authors have adequately addressed your comments raised in a previous round of review and you feel that this manuscript is now acceptable for publication, you may indicate that here to bypass the “Comments to the Author” section, enter your conflict of interest statement in the “Confidential to Editor” section, and submit your "Accept" recommendation.

Reviewer #1: All comments have been addressed

Reviewer #2: All comments have been addressed

2. Is the manuscript technically sound, and do the data support the conclusions?

Reviewer #1: Yes

Reviewer #2: Yes

3. Has the statistical analysis been performed appropriately and rigorously? 

Reviewer #1: Yes

Reviewer #2: Yes

4. Have the authors made all data underlying the findings in their manuscript fully available?

Reviewer #1: Yes

Reviewer #2: Yes

5. Is the manuscript presented in an intelligible fashion and written in standard English?

Reviewer #1: Yes

Reviewer #2: Yes

6. Review Comments to the Author

Reviewer #1: I appreciate the author's extensive revisions to the manuscript addressing all major concerns. The addition of qualitative data I think strengthens the results and provided for a much more comprehensive assessment of the subject. I enjoyed reading this finalized version and have no further comments for editing.

Reviewer #2: the author has addressed the comments provided and additional information needed to explain the selection of the study sites and sample that was key and missing in previous version of the manuscript.

7. PLOS authors have the option to publish the peer review history of their article (what does this mean?). If published, this will include your full peer review and any attached files.

Reviewer #1: No

Reviewer #2: No

---

## [Editor Report · Acceptance letter]

22 Dec 2020

PONE-D-20-24320R1 

Social networks influence farming practices and agrarian sustainability 

Dear Dr. Albizua:

I'm pleased to inform you that your manuscript has been deemed suitable for publication in PLOS ONE. Congratulations! Your manuscript is now with our production department. 

Kind regards, 

on behalf of

Dr. Sieglinde S. Snapp 

Academic Editor

PLOS ONE